# Certification for Differentially Private Prediction in Gradient-Based Training

**Matthew Wicker** [* 1 2]  **Philip Sosnin** [* 1]  **Igor Shilov** [1]  **Adrianna Janik** [3]  **Mark N. Mueller** [4 5]
**Yves-Alexandre de Montjoye** [1]  **Adrian Weller** [6]  **Calvin Tsay** [1]

## Abstract

We study private prediction where differential privacy is achieved by adding noise to the outputs of a non-private model. Existing methods rely on noise proportional to the global sensitivity of the model, often resulting in sub-optimal privacy-utility trade-offs compared to private training. We introduce a novel approach for computing dataset-specific upper bounds on prediction sensitivity by leveraging convex relaxation and bound propagation techniques. By combining these bounds with the smooth sensitivity mechanism, we significantly improve the privacy analysis of private prediction compared to global sensitivity-based approaches. Experimental results across real-world datasets in medical image classification and natural language processing demonstrate that our sensitivity bounds are can be orders of magnitude tighter than global sensitivity. Our approach provides a strong basis for the development of novel privacy preserving technologies.

## 1. Introduction

Modern machine learning systems have shown significant promise across diverse domains such as medical imaging, autonomous driving, and sentiment analysis (Bommasani et al., 2021). The potential use of such systems in situations that require the use of human data for training has led to an increase in data privacy concerns (Song et al., 2017; Carlini et al., 2023). Ensuring that machine learning models meet rigorous privacy guarantees is a necessary prerequisite for responsible deployment (Gadotti et al., 2024).

---
[*]Equal contribution [1]Department of Computing, Imperial College London, London, UK [2]The Alan Turing Institute, London, UK [3]Accenture Labs, Dublin, Ireland [4]Department of Computer Science, ETH Zurich, Zurich, Switzerland [5]LogicStar.ai, Zurich, Switzerland [6]Department of Engineering, University of Cambridge, Cambridge, UK. Correspondence to: Calvin Tsay <c.tsay@imperial.ac.uk>.

*Proceedings of the 42nd International Conference on Machine Learning*, Vancouver, Canada. PMLR 267, 2025. Copyright 2025 by the author(s).

Differential privacy (DP) has emerged as a primary tool for understanding and mitigating the leakage of private user information when deploying machine learning models (Dwork et al., 2006). Differential privacy provides a probabilistic guarantee on the amount of information an adversary can extract from model predictions or parameters (Ji et al., 2014). The most popularly deployed algorithm for achieving DP in machine learning is private training, e.g., with DP-SGD (Abadi et al., 2016), which privatizes the learning process such that the final model parameters come with a DP guarantee. Private training does come with some considerable drawbacks: privacy parameters must be fixed prior to training, leading to potentially costly training re-runs, and DP training may bias model performance in unintended (and potentially harmful) ways, e.g., with respect to discrimination (Fioretto et al., 2022).

One alternative to private training is private prediction, which achieves differential privacy by adding noise to the outputs of a non-private model (Dwork & Feldman, 2018). Unlike private training, private prediction allows users to dynamically adjust their privacy budget depending on the privilege of the user or sensitivity of the application. Additionally, private prediction can be readily applied to even the most complex training configurations such as federated learning. Despite these appealing properties, current private prediction algorithms have been observed to have an empirically unfavorable privacy-utility trade-off compared to differentially private training (van der Maaten & Hannun, 2020). Yet, recent studies have concluded that the privacy analysis given by current private prediction algorithms can be substantially improved (Chadha et al., 2024).

In this work, we present a novel framework that enables tighter privacy analysis when employing private prediction. We observe that a primary source of looseness in private prediction is the use of the *global* prediction sensitivity—the largest change in prediction between any two observable datasets—when privatizing model outputs. Our approach addresses this by using advances in non-convex optimization and bound propagation (Wicker et al., 2022) in order to compute upper-bounds on the *local* sensitivity of a given prediction. This involves computing the set of all reachable predictions when any arbitrary set of $k$ points is added or removed from the provided dataset. Using our bounds on

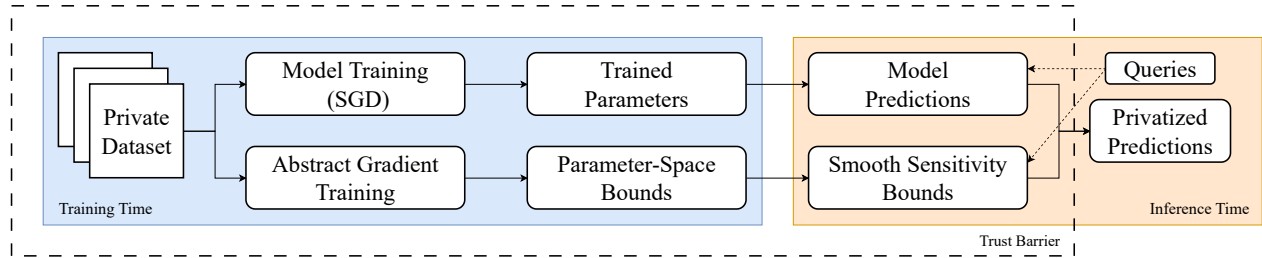

*Figure 1.* Overview of the approach (1) A model is trained using stochastic gradient descent. (2) AGT is used to compute parameter space bounds for up to $k$ arbitrary removals/additions to the dataset. (3) A certification procedure is used to bound the smooth sensitivity of a model prediction. (4) The smooth sensitivity is used to release the model prediction privately.

a prediction's local sensitivity, we show that we are able to leverage bounds from smooth sensitivity (Nissim et al., 2007) to provide tighter guarantees of differential privacy compared with the standard approach of using the global sensitivity. Empirically, we compare our approach to private prediction with private training using DP-SGD (Abadi et al., 2016) and ensemble-based private prediction approaches similar to PATE (Papernot et al., 2016). We consider a series of synthetic and real-world benchmarks including medical imaging and natural language processing. Across all tested datasets we find cases where our approach enables substantially tighter privacy analysis. We believe that this novel verification-centric framework represents a promising avenue for tightening the privacy analysis of private prediction methods and for the development of new of privacy preserving technologies. In summary this paper makes the following contributions:

- We provide a novel algorithm for bounding the reachable set of model parameters given a bound $k$ on the number of individuals that can be added/removed from the dataset. These bounds may be of independent interest to developing new privacy preserving technologies.

- We use our bound on reachable model parameters to bound the local sensitivity of any model prediction, and we prove necessary upper bounds on the smooth sensitivity to provide tighter privacy analysis.

- We validate our bounds with extensive experiments on a variety of datasets from medical imaging and sentiment classification including fully connected, convolutional, and large language models. We find that our approach offers bounds that can be orders of magnitude tighter than global sensitivity.

## 2. Related Works

DP has enabled the adoption of privacy-preserving machine learning in a variety of industries (Dwork et al., 2014), yet *post-hoc* audits have revealed a gap between attacker strength and guarantees offered by DP (Carlini et al., 2022;

Yu et al., 2022). As a result, several works seek more specific, and thus sharper, privacy guarantees. For example, Nissim et al. (2007) and Liu et al. (2022) use notions of local sensitivity to produce tighter bounds. In Ligett et al. (2017), the authors privately search the space of privacy-preserving parameters to tune performance on a given dataset, while in Yu et al. (2022) the authors propose individual differential privacy, which can compute tighter privacy bounds for given individuals in the dataset. Unlike this work, these rely solely on private training e.g., DP-SGD (Abadi et al., 2016).

We consider private prediction where one is interested in privatizing the output predictions of a model (Liu et al., 2019). While PATE's noisy softmax may be interpreted in this light (Papernot et al., 2016), private prediction has been investigated largely in the context of learning theory (Bassily et al., 2018; Nandi & Bassily, 2020). In practice, it is found that training-time privacy such as DP-SGD is preferable to prediction-time privacy (van der Maaten & Hannun, 2020). However, recent audits have discussed a significant lack of tightness in private prediction approaches (Chadha et al., 2024). Our framework presents a novel tightening of the privacy analysis of private prediction and, to the best of our knowledge, this work provides the first verification-centric approach to private prediction which will hopefully enable even more progress in this direction. We detail further related works in Appendix A.

## 3. Preliminaries

### 3.1. Differential Privacy

Differential privacy has become a widely recognized standard that offers robust privacy guarantees for algorithms that analyze databases.

**Definition 3.1** (($\epsilon, \delta$)-Differential Privacy (Dwork et al., 2014))**.** A randomized mechanism $\mathcal{M}$ is ($\epsilon, \delta$)-differentially private if, for all pairs of adjacent datasets $x, y \in \mathcal{D}$ and any $S \subseteq \text{Range}(\mathcal{M})$,

$$\mathbb{P}\big(\mathcal{M}(x) \in S\big) \leq e^{\epsilon}\mathbb{P}\big(\mathcal{M}(y) \in S\big) + \delta \qquad (1)$$

In this work, we define distances between datasets using the Hamming distance $d(x, y)$, which equals the number of entries in which $x$ and $y$ differ. Adjacent datasets are those where $d(x, y) = 1$. The parameter $\epsilon$, known as the privacy budget, controls the privacy loss of $\mathcal{M}$.

A deterministic query $f$ on a database $x \in \mathcal{D}$ can be made differentially private by adding noise calibrated to the sensitivity of the function.

**Definition 3.2** (Global Sensitivity (Dwork et al., 2014)). A function $f : \mathcal{D} \rightarrow \mathbb{R}^n$ has global ($\ell_1$) sensitivity

$$\text{GS}(f) = \max_{x,y:d(x,y) \leq 1} \|f(x) - f(y)\|_1 \qquad (2)$$

Releasing the result of the query plus additive noise drawn from a Laplace distribution with scale $\text{GS}(f)/\epsilon$ satisfies $(\epsilon, 0)$ differential privacy. Although this mechanism ensures differential privacy, the global sensitivity represents the worst case sensitivity over all datasets, which may not reflect the function's sensitivity at a particular instance. Alternative measures of sensitivity have been proposed, such as the local sensitivity:

**Definition 3.3** (Local Sensitivity (Dwork et al., 2014)). For $f : \mathcal{D} \rightarrow \mathbb{R}^n$, the local sensitivity at a point $x \in \mathcal{D}$ is

$$\text{LS}(f, x) = \max_{y:d(x,y) \leq 1} \|f(x) - f(y)\|_1 \qquad (3)$$

Unfortunately, the local sensitivity is not itself a private quantity and cannot be used directly to achieve differential privacy. However, Nissim et al. (2007) proposed a smooth upper bound on the local sensitivity that can be used to calibrate noise while satisfying differential privacy. Formally, the *maximum local sensitivity* at a distance $k$ is:

**Definition 3.4** (Maximum Local Sensitivity, (Nissim et al., 2007)). The maximum local sensitivity at a distance $k$ is given by

$$A^k(f, x) = \max_{y:d(x,y) \leq k} \text{LS}(f, y) \qquad (4)$$

We can now define the smooth sensitivity in terms of $A^k(f, x)$:

**Definition 3.5** (Smooth Sensitivity, (Nissim et al., 2007)). The $\beta$-smooth sensitivity of a function $f$ at a point $x \in \mathcal{D}$ is

$$\text{SS}^\beta(f, x) = \max_{k \in \mathbb{N}^+} e^{-\beta k} A^k(f, x) \qquad (5)$$

Taking directly from Nissim et al. (2007), the randomized algorithm that returns $f(x) + \text{Cauchy}\left(6\,\text{SS}^\beta(f, x)/\epsilon\right)$ is $(\epsilon, 0)$-differentially private[1] for $\beta \leq \epsilon/6$.

---

[1]For a 1-dimensional query $f$.

## 3.2. Private Prediction

The private prediction setting focuses on ensuring the privacy of a machine learning model's predictions. Specifically, this approach assumes that a potential adversary does not have direct access to the model itself but is limited to making a fixed number of queries, $Q$, to the model.

**Notation.** We denote a machine learning model as a parametric function $f^\theta : \mathbb{R}^n \rightarrow \mathcal{Y}$ with parameters $\theta \in \Theta$, which maps from features $x \in \mathbb{R}^n$ to labels $y \in \mathcal{Y}$. We consider supervised learning in the classification setting with a labeled dataset $D = \{(x^{(i)}, y^{(i)})\}_{i=1}^N$. The model parameters are trained, starting from some initialization $\theta'$, using a gradient-based algorithm, denoted as $M$, as $\theta = M(f, \theta', D)$. In other words, given a model, initialization, and dataset, the training function $M$ returns the "trained" parameters $\theta$. In this paper, we consider only the binary classification setting of $\mathcal{Y} = \{0, 1\}$, though the results can be generalized to the multi-class setting.

For a given query point $x$, the private prediction setting is concerned with releasing the prediction $f^{M(f,D,\theta')}(x)$ while satisfying Definition 3.1. For ease of exposition, we will use the shorthand $f_x(D) = f^{M(f,D,\theta')}(x)$ to denote the prediction of the model at a point $x$, where differential privacy must be ensured with respect to the training dataset $D$.

**Prediction Sensitivity.** To release the predictions of the model while preserving privacy, one can employ the following response mechanism. We assume a no-box setting, i.e., we take the output of the model, $f_x(D)$, to be a binary label of the model's prediction. To privatize the response, one releases

$$R(x) = \begin{cases} 1 & \text{if } f_x(D) + \text{Lap}(1/\epsilon) > 0.5, \\ 0 & \text{otherwise.} \end{cases} \qquad (6)$$

This mechanism satisfies Definition 3.1, as the global sensitivity $GS(f_x)$ is equal to 1.

**Subsample-and-Aggregate.** Subsample-and-aggregate mechanisms, such as the one employed by PATE (Papernot et al., 2016), are the current state-of-the-art in private prediction. In this setting, the data are partitioned into $T$ disjoint subsets, with $T$ models $\{f^{(i)}\}_{i=1}^T$, trained separately on each subset. The models are then deployed as an ensemble voting classifier $g$ with the following private aggregation mechanism,

$$g(x) = \arg\max_j \left\{ n_j(x) + \text{Lap}\left(\frac{2}{\epsilon}\right) \right\} \qquad (7)$$

where $n_j(x) = |\{i : i \in [T], f^{(i)}(x) = j\}|$ are the label counts for a class $j$. This response mechanism satisfies $(\epsilon, 0)$-differential privacy.

**Composition.** For a given inference budget of $Q$ queries, the total privacy loss can be computed using composition theorems (Dwork et al., 2014). Standard composition states that the repeated application of $Q$ $(\epsilon, \delta)$-differentially private queries satisfies $(Q\epsilon, Q\delta)$-differential privacy. Advanced composition allows the same queries to satisfy $(\epsilon', \delta' + Q\delta)$ differential privacy, where $\delta' > 0$ and $\epsilon' = \sqrt{2Q \ln(1/\delta')}\epsilon + Q\epsilon(e^\epsilon - 1)$. When $Q$ is small, standard composition may yield a smaller privacy loss than advanced composition. In our experiments, we choose the composition theorem that results in the smallest privacy loss.

## 4. Methodology and Computations

In this section, we outline a novel framework, termed Abstract Gradient Training (AGT), for bounding the sensitivity of predictions made by a trained machine learning model. Proofs of all theorems are given in Appendix D. In particular, AGT uses reachability analysis of the training process to certify the following property:

**Definition 4.1** (Prediction Stability). Let $f$ be a machine learning model trained on a dataset $D$ and queried at a point $x$. The prediction $f_x(D)$ is said to be stable at a distance $k$ if $\|f_x(D) - f_x(D')\|_1 = 0$ for all datasets $D'$ with $d(D, D') \leq k$.

Intuitively, this means that adding or removing up to $k$ entries in the training dataset cannot change the outcome of the prediction at the point $x$. Any distance $k$ for which a prediction is stable implies the following result.

**Lemma 4.2.** *For any $k$ satisfying Definition 4.1, the maximum local sensitivity at a distance of $k - 1$ is zero. That is, $A^{k-1}(f_x, D) = 0$.*

In the remainder of this section, we first introduce the notion of valid parameter space bounds, and show how they can be used to certify whether a given distance $k$ satisfies Definition 4.1. We then present an algorithmic framework for efficiently computing parameter-space bounds. In Section 5, we show how these results can be used to improve privacy accounting or performance in the private prediction setting.

### 4.1. Certification of Prediction Stability via Parameter Space Bounds

Determining whether a prediction from a general machine learning model satisfies Definition 4.1 is a non-trivial task and may be computationally intractable for many classes of models. Instead, our focus is on *soundly* answering the question "Is the prediction $f_x(D)$ stable at a distance $k$?". Importantly, this approach allows us only to certify that the property holds for a given $k$; it does not enable us to conclude that the property fails to hold when certification is not possible.

Formally, we wish to upper-bound the following optimization problem:

$$\max_{D'} \|f_x(D) - f_x(D')\|_1 \quad \text{s.t. } d(D, D') \leq k \quad (8)$$

If the above problem is upper bounded by 0, then we can conclude that the prediction is stable at a distance $k$. However, optimizing over the space of all datasets $D'$ is practically intractable. Rather than working in the space of datasets, we introduce the concept of *valid parameter-space bounds*, which enables efficient certification of the above optimization problem.

**Definition 4.3** (Valid Parameter-Space Bounds). Let $f$ be a model trained on a dataset $D$ via a training algorithm $M$ with parameter initialization $\theta'$. A domain $T^k \subseteq \Theta$ is said to be a valid parameter-space bound at a distance $k$ if and only if: $M(f, \theta', D') \in T^k$ for all $D' : d(D, D') \leq k$.

This definition requires that the parameters obtained by training on a dataset $D'$, located at a distance $\leq k$ from $D$, must lie within the set $T^k$. However, the converse does not hold, as $T^k$ may over-approximate the true reachable parameter space.

Using this definition we can shift the certification from relying on computations over the set of all perturbed datasets to being over an (over-approximated) parameter-space domain $T^k$. In particular we have:

**Lemma 4.4.** *For any valid parameter-space bounds $T^k$ satisfying Definition 4.3, proving that $\forall \theta \in T^k$, $\|f^\theta(x) - f^{M(f,\theta',D)}(x)\|_1 = 0$ suffices as proof that $\forall D' : d(D, D') \leq k$, $\|f^{M(f,\theta',D')}(x) - f^{M(f,\theta',D)}(x)\|_1 = 0$.*

While it is infeasible to bound (8) by working in the space of datasets, Lemma 4.4 defines corresponding properties that can be bounded straightforwardly starting with a parameter set $T$. Certifying for a given $x$ that $f^\theta(x)$ is constant for any $\theta \in T$ is discussed in more detail in Section 4.3.

### 4.2. Computing Valid Parameter-Space Bounds

This section presents the core of our framework: an algorithm that can compute valid parameter space bounds for any gradient-based training algorithm[2]. We call this algorithm *Abstract Gradient Training* and present it in Algorithm 1.

We start by introducing two assumptions. First, for expositional purposes, we assume that parameter bounds $T^k$ take the form of an interval: $[\theta_L, \theta_U]$ s.t. $\forall i, [\theta_L]_i \leq [\theta_U]_i$. This can be relaxed to linear constraints for more expressive parameter bounds at the cost of increased computational complexity. Second, we assume that the parameter initialization $\theta'$ and data ordering are both arbitrary, but fixed. The

---

[2] Here we present AGT only for stochastic gradient descent, but our framework is applicable to other first-order training procedures, such as those based on momentum.

---

**Algorithm 1** ABSTRACT GRADIENT TRAINING FOR COMPUTING VALID PARAMETER-SPACE BOUNDS

---

1: **input:** $f$ - model, $\theta'$ - init. params., $D$ - dataset, $E$ - epochs, $\alpha$ - learning rate, $k$ - # of additions/removals, $\gamma$ - clipping parameter.
2: **output:** $\theta$ - nominal SGD parameter, $[\theta_L, \theta_U]$ - valid parameter space bound for up to $k$ additions/removals.
3: $\theta \leftarrow \theta'$; $[\theta_L, \theta_U] \leftarrow [\theta', \theta']$          *// Initialize nominal parameter and interval bounds.*
4: **for** $E$-many epochs **do**
5:     **for** each batch $B \subset D$ **do**
6:        $\Delta\theta \leftarrow \frac{1}{|B|} \sum_{(x,y) \in B} \text{Clip}_\gamma \left[ \nabla_\theta \mathcal{L} \left( f^\theta(x), y \right) \right]$      *// Compute the nominal SGD parameter update.*
7:        $\theta \leftarrow \theta - \alpha\Delta\theta$         *// Update the nominal parameter.*
8:        $\Delta\Theta \leftarrow \left\{ \frac{1}{|\widetilde{B}|} \sum_{(\tilde{x}, \tilde{y}) \in \widetilde{B}} \text{Clip}_\gamma \left[ \nabla_{\tilde{\theta}} \mathcal{L} \left( f^{\tilde{\theta}}(\tilde{x}), \tilde{y} \right) \right] \mid d(\widetilde{B}, B) \leq k, \tilde{\theta} \in [\theta_L, \theta_U] \right\}$    *// Define the set of descent directions.*
9:        Compute $\Delta\theta_L, \Delta\theta_U$ s.t. $\Delta\theta_L \preceq \Delta\theta \preceq \Delta\theta_U \quad \forall \Delta\theta \in \Delta\Theta$    *// Compute bounds on the possible descent directions.*
10:       $\theta_L \leftarrow \theta_L - \alpha\Delta\theta_U$;    $\theta_U \leftarrow \theta_U - \alpha\Delta\theta_L$     *// Update the reachable parameter interval.*
11:     **end for**
12: **end for**
13: **return** $\theta, [\theta_L, \theta_U]$

---

latter assumption on data ordering is purely an expositional convenience that is relaxed in Appendix D.3.

Algorithm 1 proceeds by soundly bounding the effect of the worst-case removals and/or additions for each batch encountered during training. We therefore have the following theorem, which is proved in Appendix D:

**Theorem 4.5.** *Given a model $f$, dataset $D$, arbitrary but fixed initialization, $\theta'$, bound on the number of added or removed individuals, $k$, and learning hyper-parameters including: batch size $b$, number of epochs $E$, and learning rate $\alpha$, Algorithm 1 returns valid parameter-space bounds on the stochastic gradient training algorithm, $M$, that satisfy Definition 4.3.*

We note that the clipping procedure $\text{Clip}_\gamma$ in Algorithm 1 is a truncation operator that clamps all elements of its input to be between $-\gamma$ and $\gamma$, while leaving those within the range unchanged. This is distinct from the $\ell_2$-norm clipping typically employed by privacy-preserving mechanisms such as DP-SGD (Abadi et al., 2016). In this work, we chose to use the truncation operator as it is more amenable to bound-propagation.

The set $\Delta\Theta = \{\cdot \mid d(\widetilde{B}, B) \leq k, \tilde{\theta} \in [\theta_L, \theta_U]\}$ (line 8) represents the set of all possible descent directions that can be reached at this iteration under up to $k$ additions or removals from each batch. In particular, $\tilde{\theta}$ accounts for any $k$ removals from and/or additions to each *previously seen* batch (via valid parameter-space bounds), while $\widetilde{B}$ represents the effect of $k$ removals/additions from the *current* batch. Computing this set exactly is not tractable, so we instead compute an element-wise, over-approximated, lower and upper bound $\Delta\theta_L, \Delta\theta_U$ using the procedure in Theorem 4.6. These bounds are then combined with $\theta_L, \theta_U$ using sound interval arithmetic to produce a new valid parameter-space bound.

**Theorem 4.6** (Bounding the descent direction). *Given a nominal batch $B = \left\{ \left( x^{(i)}, y^{(i)} \right) \right\}_{i=1}^{b}$ with batch size $b$, a parameter set $[\theta_L, \theta_U]$ and clipping level $\gamma$, the parameter update vector*

$$\Delta\theta = \frac{1}{|\widetilde{B}|} \sum_{\left(\tilde{x}^{(i)}, \tilde{y}^{(i)}\right) \in \widetilde{B}} \text{Clip}_\gamma \left[ \nabla_\theta \mathcal{L} \left( f^\theta(\tilde{x}^{(i)}), \tilde{y}^{(i)} \right) \right]$$

*is bounded element-wise by*

$$\Delta\theta_L = \frac{1}{b} \left( \underset{b-k}{\text{SEMin}} \left\{ \delta_L^{(i)} \right\}_{i=1}^{b} - k\gamma \mathbf{1}_d \right)$$
$$\Delta\theta_U = \frac{1}{b} \left( \underset{b-k}{\text{SEMax}} \left\{ \delta_U^{(i)} \right\}_{i=1}^{b} + k\gamma \mathbf{1}_d \right)$$

*for any perturbed batch $\widetilde{B}$ derived from $B$ by adding up to $k$ and removing up to $k$ data-points. The terms $\delta_L^{(i)}, \delta_U^{(i)}$ are sound bounds that account for the worst-case effect of additions/removals in any previous iterations. That is, they bound the gradient given any parameter $\theta^\star \in [\theta_L, \theta_U]$ in the reachable set, i.e. $\delta_L^{(i)} \leq \delta^{(i)} \leq \delta_U^{(i)}$ for all*

$$\delta^{(i)} \in \left\{ \text{Clip}_\gamma \left[ \nabla_{\tilde{\theta}} \mathcal{L} \left( f^{\tilde{\theta}}(x^{(i)}), y^{(i)} \right) \right] \mid \tilde{\theta} \in [\theta_L, \theta_U] \right\}.$$

The operations $\text{SEMax}_a$ and $\text{SEMin}_a$ in the above theorem correspond to taking the sum of the element-wise top/bottom-$a$ elements. These operations are discussed in more detail in Appendix D.4.

**Computing Gradient Bounds.** Many of the computations in Algorithm 1 are typical computations performed during stochastic gradient descent. However, lines 8-9 involve bounding non-convex optimization problems. In particular, bounding the descent directions using Theorem 4.6 requires bounds $\delta_L^{(i)}, \delta_U^{(i)}$ on the gradients of a machine learning model $f(x, \theta)$ w.r.t. perturbations about $\theta$. We note that

solving problems of the form $\min \{\cdot \mid x \in [x_L, x_U]\}$ has been well-studied in the context of adversarial robustness certification (Huchette et al., 2023; Tsay et al., 2021) with extensions that are applicable to our setting (Gowal et al., 2018; Wicker et al., 2020; 2022). All details of computing gradient bounds for neural network models using IBP can be found in Appendix B. This approach can also be generalized to cover non-neural network machine learning models.

### 4.3. Algorithm Analysis and Discussion

**Certification of Prediction Stability.** Lemma 4.4 establishes that once we have our parameter bounds (i.e., from Algorithm 1), we can bound (8) and therefore decide whether the given prediction is stable at a distance $k$. This is done by propagating the input $x$ through the neural network with the interval from Algorithm 1 as the networks parameters (e.g., as in Wicker et al. (2020)), which produces an interval over output space. It is then straightforward to compute the largest distance between elements of this output interval, or, in the case of classification, the largest change in the prediction, e.g., if the prediction changes (Gowal et al., 2018). Following Lemma 4.4, these computations produce an upper bound on the optimization problem (8), which we will use for tighter privacy analysis in subsequent sections.

**Computing the Maximum Stable Distance.** As we will discuss below, finding any $k$ for which Definition 4.1 holds suffices to improve the performance of private prediction mechanisms. However, the tightest privacy guarantees are obtained by computing the *largest* distance $k$ satisfying Definition 4.1. Therefore, it seems that one must run Algorithm 1 for all $k \in \{1, \ldots, |D|\}$, which incurs a significant computational cost. In practice, however, we find that the majority of the privacy benefits can be achieved with fewer than 10 runs of Algorithm 1. We explore this relationship further in Appendix C.

**Computational Complexity.** To analyze the time complexity of our algorithm in comparison to standard stochastic gradient descent (SGD), we focus on the operations described in Theorem 4.6. First, computing the gradient bounds $\delta_L^{(i)}$ and $\delta_U^{(i)}$ for each sample $i$ in the batch using interval propagation requires at most four times the cost of regular training (see Appendix B). Once the bounds are computed, selecting the top or bottom $k$ gradient bounds has a time complexity of $\mathcal{O}(b)$, where $b$ is the batch size. Thus, the time complexity for a single run of Algorithm 1 is a constant factor times the complexity of standard training.

Empirically, we observe that a single run of AGT incurs a wall-clock time that is 2–4 times that of regular training. As noted above, fewer than 10 runs of Algorithm 1 are generally sufficient to achieve most of the privacy benefits, leading to a total training and inference penalty of 20–40

times that of standard approaches.

**Limitations.** Algorithm 1 can derive valid bounds for the parameter space of any gradient-based training algorithm. However, the tightness of these bounds depends heavily on the specific architecture, hyperparameters, and training process employed. In particular, the bound propagation between consecutive iterations assumes the worst-case additions and removals at *every parameter index*. As a result, achieving meaningful guarantees with this method may necessitate training with larger batch sizes or fewer epochs than usual. Moreover, some loss functions, such as multi-class cross-entropy, result in especially loose interval relaxations. Consequently, AGT tends to provide weaker guarantees for multi-class problems compared to binary classification tasks. We anticipate that future developments in tighter bound-propagation methods, such as those leveraging more expressive abstract domains, could address these limitations.

## 5. Improved Private Prediction using Smooth Sensitivity

The previous section outlined a framework for certifying whether a prediction from a model $f$ at a point $x$ is stable for up to $k$ removals or additions to the training dataset. In this section, we outline how these certificates can be used to bound the smooth sensitivity, and improve the performance of private prediction mechanisms. Proofs of all theorems are given in Appendix D.

### 5.1. Prediction Sensitivity using Smooth Sensitivity

In this section, we describe how to use certificates of prediction stability to upper-bound the smooth sensitivity of a binary classifier $f$. In particular, we have the following result:

**Theorem 5.1.** *Let $f_x(D)$ denote the prediction of a binary classifier trained on a dataset $D$ at a point $x$. If $f_x(D)$ is stable at a distance of $k'$, then the following provides an upper bound on the $\beta$-smooth sensitivity:*

$$\mathrm{SS}^\beta(f_x, D) = \max_{k \in \mathbb{N}^+} e^{-\beta k} A^k(f_x, D) \leq e^{-\beta k'} \quad (9)$$

To establish this result, we observe that $A^k(f_x, D)$ is a monotonically increasing function of $k$ and assumes values only in $\{0, 1\}$, meaning it behaves as a step function with respect to $k$. Since the exponential term $e^{-\beta k}$ is monotonically decreasing, their product achieves its maximum at $k^\star = \min\{k : A^k(f_x, D) = 1\}$. By Lemma 4.2, if $k'$ is stable, then $k^\star > k' - 1$. Consequently, the smooth sensitivity is bounded above by $\exp(-\beta k')$. For the detailed proof of Theorem 5.1, refer to Appendix D.

Theorem 5.1 is trivially satisfied when $k' = 0$, resulting

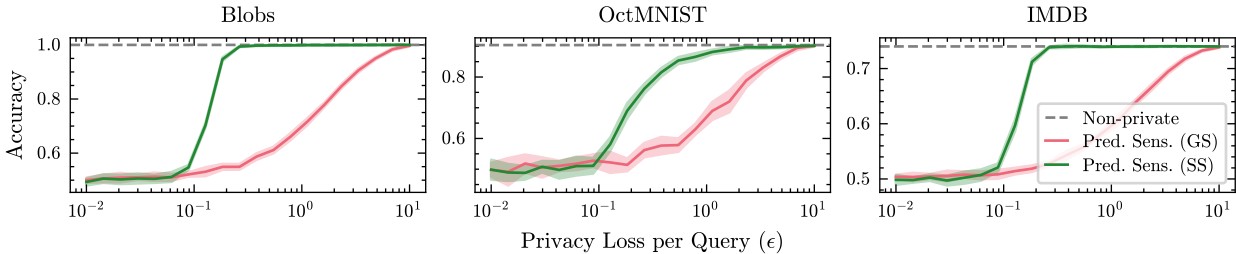

Figure 2. Private prediction accuracy for a single model using global sensitivity vs our smooth sensitivity bounds.

in $SS^\beta(f_x, D) = GS(f_x)$. However, significantly tighter bounds on the smooth sensitivity can be achieved for any prediction certified as stable at a distance $k'$ using Algorithm 1. These tighter bounds can then be leveraged to calibrate noise and release predictions through the following response mechanisms.

**Corollary 5.2.** *Suppose a prediction $f_x(D)$ is stable at a distance of $k'$. Then, the following response mechanism satisfies $(\epsilon, 0)$-differential privacy:*

$$R(x) = \begin{cases} 1 & \text{if } f_x(D) + z > 0.5, \\ 0 & \text{otherwise,} \end{cases} \quad (10)$$

*where $z \sim \text{Cauchy}\left(6\exp(-\epsilon k'/6)/\epsilon\right)$.*

When $k' \gg 1$, Corollary 5.2 enables a substantial reduction in the noise added to predictions compared to mechanisms that rely on global sensitivity.

### 5.2. Subsample and Aggregate using Smooth Sensitivity

In this section, we propose a private aggregation mechanism using the smooth sensitivity. As before, we first partition the dataset into $T$ disjoint subsets and train a binary classifier $f^{(i)}$ on each. At any query point $x$, we compute the (non-private) ensemble voting response $g$, defined as:

$$g(x) = \begin{cases} 1 & \text{if } n_1(x) \geq n_0(x) \\ 0 & \text{otherwise} \end{cases} \quad (11)$$

where $n_j(x) = |\{i : i \in [T], f^{(i)}(x) = j\}|$ are the label counts for a class $j$.

Consider the stable distance of the response $g$. In particular, we note that at least $n = \left\lceil \frac{|n_1(x) - n_0(x)|}{2} \right\rceil$ votes must be flipped to cause the overall ensemble response $g$ to change. Suppose that each prediction $f^{(i)}(x)$ is stable up to some distance $k^{(i)}$. Then, the number of entries in the original dataset $D$ that must be changed to cause $n$ votes to flip is at least the sum of the $n$ smallest stable distances $k^{(i)}$. Formally, we have the following result:

**Theorem 5.3.** *Consider an ensemble comprising $T$ classifiers $f^{(i)}$, $i = 1, \ldots, T$, each trained on a disjoint subset*

*of the dataset $D$. Let $g_x$ denote the (non-private) ensemble aggregation function as defined in (11), queried at a point $x$. If the prediction of each classifier $f^{(i)}(x)$ is stable at a distance of $k^{(i)}$, then:*

$$A^{K-1}(g_x, D) = 0, \quad (12)$$

*where $K = \sum_{i \in S_n} k^{(i)} + n - 1$ and $S_n$ is the set of indices corresponding to the $n$ classifiers with the smallest stable distances $k^{(i)}$. Here, $n = \left\lceil \frac{|n_1(x) - n_0(x)|}{2} \right\rceil$ is half the distance between the vote counts.*

Combining Corollary 5.2 and Theorem 5.3, we find that we can privatize the ensemble response function (11) adding noise drawn from a $\text{Cauchy}(6\exp(-\epsilon K/6)/\epsilon)$ distribution.

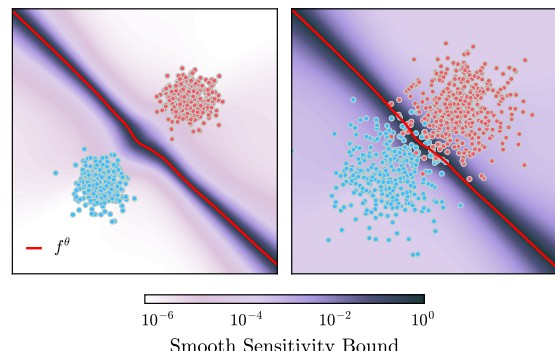

Figure 3. We use the "blobs" dataset to visualize our smooth sensitivity bounds for $\epsilon = 1.0$. The red line shows the model's decision boundary.

## 6. Experiments

In this section, we present experimental validation of our proposed private prediction mechanisms. Comprehensive details on datasets, models, training configurations, and additional results can be found in Appendix E. We evaluate our approach across three binary classification tasks:
*Blobs* – Training a logistic regression on a "blobs" dataset generated from isotropic Gaussian distributions.
*Medical Imaging* – Fine-tuning the final dense layers of a convolutional neural network to distinguish an unseen

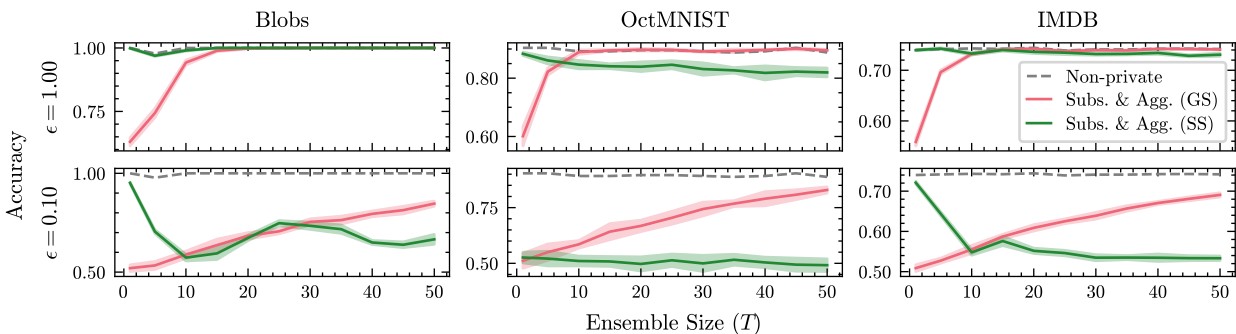

*Figure 4.* Private prediction accuracy as a function of ensemble size using global vs smooth sensitivity.

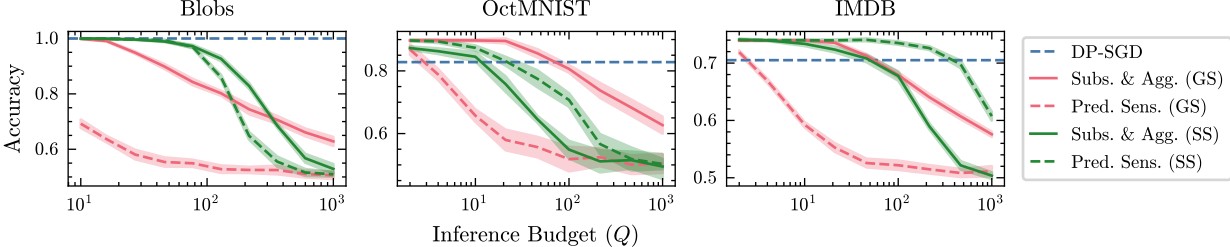

*Figure 5.* Private accuracy as a function of number of queries ($Q$) for a total privacy budget of $(\epsilon, \delta) = (10.0, 10^{-5})$. An ensemble size of $T = 25$ is used for both subsample and aggregate mechanisms.

diseased class in retinal OCT images.

*Sentiment Classification* – Training a neural network to perform sentiment analysis using GPT-2 embeddings of the IMDB movie reviews dataset.

**Smooth Sensitivity Visualization.** In Figure 3 we plot our bounds on smooth sensitivity for two different datasets: one where the data are easily separable (left panel) and one where the data are overlapping (right panel). Given that our bounds are dataset specific, we find that our bounds on smooth sensitivity generally suffer when the data are non-separable or when little data are available. We will observe these two factors throughout our experimental evaluation.

**Tightening Private Prediction in a Single Model.** Figure 2 shows the performance of the prediction sensitivity mechanism using both global sensitivity and our smooth sensitivity bounds. We observe that our bounds can maintain noise-free accuracy levels for a privacy budget $\epsilon$ up to an order of magnitude lower than the accuracy obtained using global sensitivity. The effect is most pronounced when the data are separable ("Blobs") or when the dataset contains a large number of training examples ("IMDB").

**Tightening Private Prediction in Ensembles.** We illustrate the performance of our proposed private aggregation mechanism in Figure 4. The plots show private accuracy for $\epsilon = 1.0$ (top) and $\epsilon = 0.1$ (bottom) as a function of ensemble size. For smaller ensembles ($T \leq 20$), our mechanism performs comparably to, or better than, the noisy

argmax mechanism. However, as the ensemble size grows, the amount of data available to each member decreases, resulting in Algorithm 1 yielding relatively weaker bounds for each individual member. This effect is somewhat mitigated by the summation of certified stable distances in Theorem 5.3, causing our method's performance to remain relatively stable or degrade only gradually as ensemble size increases. Future work could address this limitation by incorporating tighter bound propagation procedures or by proposing a mechanism that takes advantage of our bounds only when they are tighter than global sensitivity.

**Comparison with Private Training.** Finally, we compare the accuracy of our private prediction mechanisms to differentially private training (DP-SGD). Previous research has found that differentially private training generally achieves a better privacy-utility trade-off than private prediction (van der Maaten & Hannun, 2020). Consequently, private prediction is often reserved for Student-Teacher settings, where only a limited number of queries are made to the Teacher model to conserve the privacy budget (Papernot et al., 2016). This trend is evident in Figure 5, which depicts model accuracy as a function of inference budget for a fixed privacy loss. Private prediction performs well for a small number of queries and outperforms private training on OctMNIST and IMDB for fewer than 100 queries, but deteriorates rapidly as the number of queries increases. In the single-model setting, our smooth sensitivity bounds offer significant improvements over global sensitivity and

using them increases the maximum number by an order of magnitude compared with any other method on IMDB. Nonetheless, the highest accuracy is typically achieved with large ensembles using global sensitivity.

**Use In Teacher Models.** While we focus on the setting of issuing private predictions, our tighter bounds may be used in the semi-supervised private learning setting proposed in PATE (Papernot et al., 2016). To understand our approach in this setting we re-run the experimental setup from Figure 5, using a privacy budget of $(\epsilon, \delta) = (10, 10^{-5})$ to label $Q = 100$ data points held out from the training dataset (which we assume to be our "public" unlabeled dataset). We emphasize that training a student model does fix the privacy budget and which privacy budget is selected will have a significant effect on the results. Once this is done, resulting teacher-generated labels are then used to train a student model. Our findings indicate that the student model's performance under each mechanism aligns with the accuracy levels observed at the corresponding inference budget in Figure 5, thus confirming that our bounds are effective in this setting. We highlight that this preliminary results could be strengthened and expanded by considering the combination of our approach with other tighter accounting approaches (Papernot et al., 2018; Malek Esmaeili et al., 2021).

*Table 1.* Performance of different teacher mechanisms across datasets.

| Teacher Mechanism | Blobs | OctMNIST | IMDB |
|---|---|---|---|
| Single model, GS | 82.8 | 12.7 | 54.4 |
| Single model, SS | 99.8 | 18.7 | 73.5 |
| Subsample & agg., GS | 99.5 | 14.1 | 73.0 |
| Subsample & agg., SS | 98.1 | 19.8 | 71.7 |
| DP-SGD | 1.0 | 81.2 | 70.5 |

## 7. Conclusion

In this work, we propose a framework for computing valid bounds on a machine learning model's parameters under the addition or removal of up to $k$ data points. By certifying prediction stability, we leverage these parameter-space bounds to derive upper bounds on the smooth sensitivity of model predictions. Our results demonstrate the use of our bounds in improving the performance of private prediction. This framework represents a foundational step towards developing new certification-based techniques for privacy preserving machine learning.

**Future Directions.** While in this work we establish the use of certification to improve privacy analysis, there are many important future directions to explore including the combination of certification with tighter aggregation mechanisms (Papernot et al., 2018) or the adoption of this ap-

proach to other privacy settings (Jordon et al., 2018; Yu et al., 2022). Additionally, we highlight that advances in aggregation mechanisms or in improved certification techniques may lead to even greater improvements when used in the framework we present.

## Impact Statement

This paper presents work whose goal is to advance the field of Machine Learning. There are many potential societal consequences of our work, none which we feel must be specifically highlighted here.

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

# A. Related Works

DP has enabled the adoption of privacy-preserving machine learning in a variety of industries (Dwork et al., 2014), yet *post-hoc* audits have revealed a gap between attacker strength and guarantees offered by DP (Carlini et al., 2022; Yu et al., 2022). As a result, several works seek more specific, and thus sharper, privacy guarantees. For example, Nissim et al. (2007) and Liu et al. (2022) use notions of local sensitivity to produce tighter bounds. In (Ligett et al., 2017), the authors privately search the space of privacy-preserving parameters to tune performance on a given dataset, while in (Yu et al., 2022) the authors propose individual differential privacy, which can compute tighter privacy bounds for given individuals in the dataset. Unlike this work, these rely solely on private training e.g., DP-SGD (Abadi et al., 2016).

On the other hand, DP can additionally provide bounds in the setting of machine unlearning (Sekhari et al., 2021; Huang & Canonne, 2023). In such cases, it is more likely that guarantees are not tight due to the assumption that individuals can be both added and removed, rather than just removed (Huang & Canonne, 2023). The gold standard for unlearning (which incurs no error) is retraining. Yet, keeping the data on hand poses a privacy concern (Dwork et al., 2014), and retraining can be prohibitively costly (Nguyen et al., 2022). If one admits the privacy cost of storing and tracking all data, then retraining costs can be limited (Bourtoule et al., 2021). Existing unlearning without retraining are either restricted to linear (Guo et al., 2019) or strongly convex models (Neel et al., 2021).

The privacy setting most similar to the one adopted in this paper is the differential private prediction setting where we are interested in only privatizing the output predictions of a model (Liu et al., 2019). The PATE method may be interpreted in this light (Papernot et al., 2016), but largely this setting has been investigated in the context of learning theory (Bassily et al., 2018; Nandi & Bassily, 2020). In practice, it is found that training-time privacy such as DP-SGD is preferable to prediction-time privacy (van der Maaten & Hannun, 2020). This work can be viewed as proving tighter bounds on the private prediction setting, which allows private prediction to display some benefits over training-time privacy.

The approaches that are computationally similar to the framework established in this paper come from adversarial robustness certification (Katz et al., 2017; Gehr et al., 2018) or robust training (Gowal et al., 2018; Müller et al., 2022). These approaches typically utilize methods from formal methods (Katz et al., 2017; Wicker et al., 2018) or optimization (Huchette et al., 2023; Tsay et al., 2021). Most related to this work are strategies that provide guarantees over varying both model inputs and parameters (Wicker et al., 2020; Xu et al., 2020), as well as work on robust explanations that bound the input gradients of a model (Wicker et al., 2022). Despite some methodological relationships, none of the above methods can apply to the general training setting without the proposed framework and are unable to make statements about differential privacy.

# B. Interval Bound Propagation

In this section, we provide details of the interval bound propagation procedure required to compute the gradient bounds required by Theorem 4.6 in the context of neural network models. We define a neural network model $f^\theta : \mathbb{R}^{n_0} \to \mathbb{R}^{n_K}$ with parameters $\theta := \left\{ (W^{(i)}, b^{(i)}) \right\}_{i=1}^K$ to be a function composed of $K$ layers:

$$\hat{z}^{(k)} = W^{(k)} z^{(k-1)} + b^{(k)}, \quad z^{(k)} = \sigma\left(\hat{z}^{(k)}\right)$$

where $z^{(0)} := x$, $f^\theta(x) := \hat{z}^{(K)}$, and $\sigma$ is the activation function, which we take to be ReLU.

The standard back-propagation procedure for computing the gradients of the loss $\mathcal{L}$ w.r.t. the parameters of the neural network is given by

$$\frac{\partial \mathcal{L}}{\partial z^{(k-1)}} = \left(W^{(k)}\right)^\top \frac{\partial \mathcal{L}}{\partial \hat{z}^{(k)}}, \quad \frac{\partial \mathcal{L}}{\partial \hat{z}^{(k)}} = H\left(\hat{z}^{(k)}\right) \circ \frac{\partial \mathcal{L}}{\partial z^{(k)}}$$
$$\frac{\partial \mathcal{L}}{\partial W^{(k)}} = \frac{\partial \mathcal{L}}{\partial \hat{z}^{(k)}} \left(z^{(k-1)}\right)^\top, \quad \frac{\partial \mathcal{L}}{\partial b^{(k)}} = \frac{\partial \mathcal{L}}{\partial \hat{z}^{(k)}}$$

where $H(\cdot)$ is the Heaviside function, and $\circ$ is the Hadamard product.

**Interval Arithmetic**   Let us denote intervals over matrices as $\boldsymbol{A} := [A_L, A_U] \subseteq \mathbb{R}^{n \times m}$ such that for all $A \in \boldsymbol{A}$, $A_L \leq A \leq A_U$. We define the following interval matrix arithmetic operations:

$$\text{Addition:} \quad A + B \in [\boldsymbol{A} \oplus \boldsymbol{B}] \; \forall A \in \boldsymbol{A}, B \in \boldsymbol{B}$$

$$\text{Matrix mul.:} \quad A \times B \in [\boldsymbol{A} \otimes \boldsymbol{B}] \; \forall A \in \boldsymbol{A}, B \in \boldsymbol{B}$$

$$\text{Elementwise mul.:} \quad A \circ B \in [\boldsymbol{A} \odot \boldsymbol{B}] \; \forall A \in \boldsymbol{A}, B \in \boldsymbol{B}$$

Each of these operations can be computed using standard interval arithmetic in at most $4\times$ the computational cost of its non-interval counterpart. For example, interval matrix multiplication can be computed efficiently using Rump's algorithm (Rump, 1999). We denote interval vectors as $\boldsymbol{a} := [a_L, a_U]$ with analogous operations.

We will now describe the procedure for propagating intervals through the forward and backward passes of a neural network to compute valid gradient bounds.

**Bounding the Forward Pass**   Given these interval operations, for any input $x \in \boldsymbol{x}$ and parameters $W^{(k)} \in \boldsymbol{W}^{(k)}$, $b^{(k)} \in \boldsymbol{b}^{(k)}$, $k = 1, \ldots, K$, we can compute intervals

$$\hat{\boldsymbol{z}}^{(k)} = \boldsymbol{W}^{(k)} \otimes \boldsymbol{z}^{(k-1)} \oplus \boldsymbol{b}^{(k)}, \quad \boldsymbol{z}^{(k)} = \sigma\left(\hat{\boldsymbol{z}}^{(k)}\right)$$

such that $f^\theta(x) \in \hat{\boldsymbol{z}}^{(K)}$. The monotonic activation function $\sigma$ is applied element-wise to both the lower and upper bound of its input interval to obtain a valid output interval. Here we consider only neural networks with ReLU activations, although the interval propagation framework is applicable to many other architectures.

**Bounding the Loss Gradient**   Since we are in a classification setting, we will consider a standard cross entropy loss. Given the output logits of the neural network, $\hat{z}^{(K)} = f^\theta(x)$, the categorical cross entropy loss function is given by

$$\mathcal{L}\left(\hat{z}^{(K)}, y\right) = -\sum_i y_i \log p_i$$

where

$$p_i = \left[\sum_j \exp\left(\hat{z}_j^{(K)} - \hat{z}_i^{(K)}\right)\right]^{-1}$$

is the output of the $\mathrm{softmax}$ function and $y$ is a one-hot encoding of the true label. The gradient of the cross entropy loss $\mathcal{L}$ with respect to $\hat{z}^{(K)}$ is given by

$$\frac{\partial \mathcal{L}\left(\hat{z}^{(K)}, y\right)}{\partial \hat{z}^{(K)}} = p - y$$

The output of $p = \mathrm{softmax}(\hat{z}^{(K)})$ given any $\hat{z}^{(K)} \in \left[\hat{z}_L^{(K)}, \hat{z}_U^{(K)}\right]$ is bounded by

$$[p_L]_i = \left[\sum_j \exp\left(\left[\hat{z}_U^{(K)}\right]_j - \left[\hat{z}_L^{(K)}\right]_i\right)\right]^{-1},$$

$$[p_L]_i = \left[\sum_j \exp\left(\left[\hat{z}_L^{(K)}\right]_j - \left[\hat{z}_U^{(K)}\right]_i\right)\right]^{-1}.$$

Therefore, an interval over the gradient of the loss $\mathcal{L}$ with respect to $\hat{z}^{(K)}$ is given by

$$\frac{\partial \mathcal{L}\left(\hat{z}, y\right)}{\partial \hat{z}} = [p_L - y, p_U - y]$$

**Bounding the Backward Pass** Wicker et al. (2022) use interval arithmetic to bound derivatives of the form $\partial \mathcal{L} / \partial z^{(k)}$ and here we extend this to additionally compute bounds on the derivatives w.r.t. the parameters. Specifically, we can back-propagate $\partial \mathcal{L} / \partial \hat{z}^{(K)}$ (computed above) to obtain

$$
\begin{aligned}
\frac{\partial \mathcal{L}}{\partial z^{(k-1)}} &= \left( W^{(k)} \right)^{\top} \otimes \frac{\partial \mathcal{L}}{\partial \hat{z}^{(k)}} \\
\frac{\partial \mathcal{L}}{\partial \hat{z}^{(k)}} &= H\left( \hat{z}^{(k)} \right) \odot \frac{\partial \mathcal{L}}{\partial z^{(k)}} \\
\frac{\partial \mathcal{L}}{\partial W^{(k)}} &= \frac{\partial \mathcal{L}}{\partial \hat{z}^{(k)}} \otimes \left( z^{(k-1)} \right)^{\top} \\
\frac{\partial \mathcal{L}}{\partial b^{(k)}} &= \frac{\partial \mathcal{L}}{\partial \hat{z}^{(k)}}
\end{aligned}
$$

where $H(\cdot)$ applies the Heaviside function to both the lower and upper bounds of the interval, and $\circ$ is the Hadamard product. The resulting intervals are valid bounds for each partial derivative, that is

$$
\frac{\partial \mathcal{L}}{\partial W^{(k)}} \in \frac{\partial \mathcal{L}}{\partial W^{(k)}}, \frac{\partial \mathcal{L}}{\partial b^{(k)}} \in \frac{\partial \mathcal{L}}{\partial b^{(k)}}
$$

for all $W^{(k)} \in W^{(k)}, b^{(k)} \in b^{(k)}$ and $k = 1, \ldots, K$.

## C. Bounding the Smooth Sensitivity

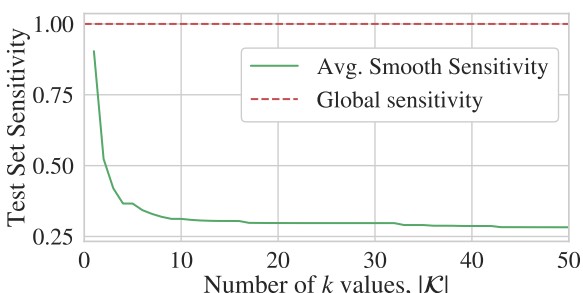

*Figure 6.* Effect of $|\mathcal{K}|$ on the tightness of the smooth sensitivity bound for the GTP-2 sentiment classification task ($\epsilon = 1.0$).

Theorem 5.1 provides a procedure for bounding the smooth sensitivity of a binary classifier $f$ queried at a point $x$. Specifically, any $k'$ for which the prediction $f_x(D)$ is certified to be stable provides an upper bound on the smooth sensitivity $SS^{\beta}(f_x, D) \leq e^{-\beta k'}$. It is clear that the tightest upper bound is obtained by certifying the prediction to be stable for the largest possible $k'$. Therefore, we employ the following procedure for finding the largest stable $k$ using AGT:

1. Choose a set of values $\mathcal{K} = \{k_i\}_{i=1}^M$ and compute the corresponding parameter-space bounds using Algorithm 1.

2. At a query point $x$, find the largest $k' \in \mathcal{K}$ for which $f_x(D)$ is stable (e.g. via binary or linear search).

The set $\mathcal{K}$ should be chosen according to the available computational budget. A more fine-grained set of $k$'s will achieve tighter sensitivity bounds, as $k'$ will fall closer to $k^{\star}$, on average. On the other hand, increasing the number of $k$ values increases the computational complexity at both training time (running Algorithm 1 for each $k$) and at inference time (finding $k'$ via binary search).

In practice, the set $\mathcal{K}$ should be chosen with greater density for smaller values of $k$, as the improvement in the tightness of the bound is more pronounced for lower values of $k$ due to the exponential decay. Specifically, going from $k' = 1$ to $k' = 2$ results in a significantly greater improvement than, for example, refining the bound between $k' = 51$ and $k' = 52$.

Furthermore, for sufficiently large values of $k$, it is often the case that the resulting bounds do not certify any points within the domain, rendering the inclusion of such large $k$ values unnecessary. Consequently, it is advantageous to prioritize smaller

$k$ values in the set $\mathcal{K}$, where the improvement in the bound is substantial, while avoiding the inclusion of excessively large $k$ values that offer diminishing returns both in terms of bound improvement and computational efficiency.

Figure 6 illustrates how increasing the number of $k$ values affects the tightness of our bounds. Even when $|\mathcal{K}| = 1$ (i.e., AGT is run for a single value of $k$), the smooth sensitivity bound is already tighter than the global sensitivity. As the number of $k$ values increases, the smooth sensitivity bound becomes progressively tighter. However, after $|\mathcal{K}| \approx 10$, the gains diminish, indicating that running AGT for a small number of $k$ values is sufficient to capture most of the privacy benefits.

# D. Proofs

### D.1. Proof of Lemma 4.2

*Proof.* Suppose $A^{k-1}(f_x, D) > 0$. Then, by the definition of $A^{k-1}$, $\exists Y : d(D, Y) \leq k - 1$ with $\mathrm{LS}(f_x, Y) > 0$. This implies

$$\exists Y, Y' : d(Y, Y') \leq 1, \|f_x(Y) - f_x(Y')\|_1 > 0. \tag{13}$$

By the triangle inequality, we have

$$d(D, Y') \leq d(D, Y) + d(Y, Y') \leq k. \tag{14}$$

Since both $Y, Y'$ are a distance $\leq k$ from $D$, we have that $f_x(D) = f_x(Y) = f_x(Y')$ by Definition 4.1. Therefore, $\|f_x(Y) - f_x(Y')\|_1 = 0$, which is a contradiction. $\square$

### D.2. Proof of Lemma 4.4

*Proof.* Let

$$\forall \theta \in T^k, \|f^\theta(x) - f^{M(f,\theta',D)}(x)\|_1 = 0. \tag{15}$$

Now, suppose there exists a dataset $D'$ such that $d(D, D') \leq k$ and $\|f^{M(f,\theta',D')}(x) - f^{M(f,\theta',D)}(x)\|_1 > 0$. By Definition 4.3, the parameters of the model trained on $D'$ must lie within the valid parameter space bounds $T^k$, i.e. $M(f, \theta', D') \in T^k$. However, this would contradict (15), as it requires $\|f^\theta(x) - f^{M(f,\theta',D)}\|_1 = 0$ for all $\theta \in T^k$. Therefore, (15) suffices as a proof that $\|f^{M(f,\theta',D')}(x) - f^{M(f,\theta',D)}(x)\|_1 = 0$ for all $D' : d(D, D') \leq k$. $\square$

### D.3. Proof of Theorem 4.5

Here we provide a proof of correctness for our algorithm (i.e., proof of Theorem 4.5) as well as a detailed discussion of the operations therein.

First, we recall the definition of valid parameter space bounds (Definition 4.3 in the main text):

$$\theta_i^L \leq \min_{D' \in T(D)} M(f, \theta', D')_i \leq M(f, \theta', D)_i \leq \max_{D' \in T(D)} M(f, \theta', D')_i \leq \theta_i^U \tag{16}$$

As well as the iterative equations for stochastic gradient descent:

$$\theta \leftarrow \theta - \alpha \Delta\theta, \qquad \Delta\theta \leftarrow \frac{1}{|B|} \sum_{(x,y) \in B} \nabla_\theta \mathcal{L}\left(f^\theta(x), y\right) \tag{17}$$

For ease of notation, we assume a fixed data ordering (one may always take the element-wise maximums/minimums over the entire dataset rather than each batch to relax this assumption).

Now, we proceed to prove by induction that Algorithm 1 maintains valid parameter space bounds on each step of gradient descent. We start with the base case of $\theta^L = \theta^U = \theta'$ according to line 1, which are valid parameter-space bounds. Our inductive hypothesis is that, given valid parameter space bounds satisfying Definition 4.3, each iteration of Algorithm 1 (lines 4–8) produces a new $\theta^L$ and $\theta^U$ that satisfy also Definition 4.3.

First, we observe that lines 4–5 simply compute the normal forward pass. Second, we note that lines 6–7 compute valid bounds on the descent direction for all possible poisoning attacks within $T(D)$. In other words, the inequality $\Delta\theta^L \leq \Delta\theta \leq \Delta\theta^U$ holds element-wise for any possible batch $\tilde{B} \in T(D)$. Combining this largest and smallest possible update with the smallest and largest previous parameters yields the following bounds:

$$\theta^L - \alpha \Delta\theta^U \leq \theta - \alpha \Delta\theta \leq \theta^U - \alpha \Delta\theta^L$$

which, by definition, constitute valid parameter-space bounds and, given that these bounds are exactly those in Algorithm 1, we have that Algorithm 1 provides valid parameter space bounds as desired. □

### D.4. Proof of Theorem 4.6

*Proof.* The nominal clipped descent direction for a parameter $\theta$ is the averaged, clipped gradient over a training batch $B$, defined as

$$\Delta\theta = \frac{1}{b} \sum_{i=1}^{b} \text{Clip}_\gamma \left[ \delta^{(i)} \right]$$

where each gradient term is given by $\delta^{(i)} = \nabla_\theta \mathcal{L} \left( f^\theta \left( x^{(i)} \right), y^{(i)} \right)$. Our goal is to bound this descent direction for the case when (up to) $k$ points are removed or added to the training data, for any $\theta \in [\theta_L, \theta_U]$. We begin by bounding the descent direction for a fixed, scalar $\theta$, then generalize to all $\theta \in [\theta_L, \theta_U]$ and to the multi-dimensional case (i.e., multiple parameters). We present only the upper bounds here; analogous results apply for the lower bounds.

**Bounding the descent direction for a fixed, scalar $\theta$.** Consider the effect of removing up to $k$ data points from batch $B$. Without loss of generality, assume the gradient terms are sorted in descending order, i.e., $\delta^{(1)} \geq \delta^{(2)} \geq \cdots \geq \delta^{(b)}$. Then, the average clipped gradient over all points can be bounded above by the average over the largest $b - k$ terms:

$$\Delta\theta = \frac{1}{b} \sum_{i=1}^{b} \text{Clip}_\gamma \left[ \delta^{(i)} \right] \leq \frac{1}{b-k} \sum_{i=1}^{b-k} \text{Clip}_\gamma \left[ \delta^{(i)} \right]$$

This bound corresponds to removing the $k$ points with the smallest gradients.

Next, consider adding $k$ arbitrary points to the training batch. Since each added point contributes at most $\gamma$ due to clipping, the descent direction with up to $k$ removals and $k$ additions is bounded by

$$\frac{1}{b} \sum_{i=1}^{b} \text{Clip}_\gamma \left[ \delta^{(i)} \right] \leq \frac{1}{b-k} \sum_{i=1}^{b-k} \text{Clip}_\gamma \left[ \delta^{(i)} \right] \leq \frac{1}{b} \left( k\gamma + \sum_{i=1}^{b} \text{Clip}_\gamma \left[ \delta^{(i)} \right] \right)$$

where the bound now accounts for replacing the $j$ smallest gradient terms with the maximum possible value of $\gamma$ from the added samples.

**Bounding the effect of a variable parameter interval.** We extend this bound to any $\theta \in [\theta_L, \theta_U]$. Assume the existence of upper bounds $\delta_U^{(i)}$ on the clipped gradients for each data point over the interval, such that

$$\delta_U^{(i)} \geq \text{Clip}_\gamma \left[ \nabla_{\theta'} \mathcal{L} \left( f^{\theta'}(x^{(i)}), y^{(i)} \right) \right] \quad \forall \theta' \in [\theta_L, \theta_U].$$

Then, using these upper bounds, we further bound $\Delta\theta$ as

$$\Delta\theta \leq \frac{1}{b} \left( k\gamma + \sum_{i=1}^{b} \text{Clip}_\gamma \left[ \delta_U^{(i)} \right] \right)$$

where, as before, we assume $\delta_U^{(i)}$ are indexed in descending order.

**Extending to the multi-dimensional case.** To generalize to the multi-dimensional case, we apply the above bound component-wise. Since gradients are not necessarily ordered for each parameter component, we introduce the $\text{SEMax}_n$ operator, which selects and sums the largest $n$ terms at each index. This yields the following bound on the descent direction:

$$\Delta\theta \leq \frac{1}{b} \left( \text{SEMax}_{b-k} \left\{ \delta_U^{(i)} \right\}_{i=1}^{b} + k\gamma \mathbf{1}_d \right)$$

which holds for any $\theta \in [\theta_L, \theta_U]$ and up to $k$ removed and replaced points.

We have established the upper bound on the descent direction. The corresponding lower bound can be derived by reversing the inequalities and substituting SEMax with the analogous minimization operator, SEMin. □

## D.5. Proof of Theorem 5.1

*Proof.* We start by noting the following properties of the maximum local sensitivity of the prediction of a binary classifier $f$ at a point $x$:

$$A^k(f_x, D) = \max_{D':d(D,D')\leq k} \text{LS}(f_x, D') \tag{18}$$

- $A^k(f_x, D)$ takes only values in $\{0, 1\}$, since $|f_x(D) - f_x(D')| \in \{0, 1\}$.

- $A^k(f_x, D)$ is monotonically increasing in $k$.

Therefore, $A^k(f_x, D)$ takes the form of a step function:

$$A^k(f_x, D) = \begin{cases} 1 & \text{if } k \geq k^\star, \\ 0 & \text{otherwise,} \end{cases} \tag{19}$$

for some $k^\star = \min\{k : A^k(f_x, D) = 1\}$. We now relate this to the definition of $\beta$-smooth sensitivity.

$$\text{SS}^\beta(f_x, D) = \max_{k \in \mathbb{N}+} e^{-\beta k} A^k(f_x, D) = \max_{k \in \mathbb{N}+} s(k) \tag{20}$$

where

$$s(k) = \begin{cases} e^{-\beta k} & \text{if } k \geq k^\star, \\ 0 & \text{otherwise.} \end{cases} \tag{21}$$

Note that the exponential term is monotonically decreasing in $k$ since $\beta > 0$. Therefore, the maximum value of $s(k)$ is attained at $k^\star$, giving $\text{SS}^\beta(f_x, D) = e^{-\beta k^\star}$. Now, we have that

$$\forall \hat{k} : \hat{k} \leq k^\star, \quad \text{SS}^\beta(f_x, D) = e^{-\beta k^\star} \leq e^{-\beta \hat{k}} \tag{22}$$

Suppose the prediction $f_x(D)$ is stable at a distance $k'$. By Lemma 4.2, this tells us that $A^{k'-1}(f_x, D) = 0$. Therefore, $k^\star > k' - 1 \Rightarrow k^\star \geq k'$. By (22), this gives us the following valid upper bound

$$\text{SS}^\beta(f_x, D) = e^{-\beta k^\star} \leq e^{-\beta k'}. \tag{23}$$

$\square$

## D.6. Proof of Corollary 5.2

*Proof.* We recall from Section 3 the following response mechanism for a 1-dimensional query $f_x(D)$:

If $\beta \leq \epsilon/6$, the algorithm that returns $f_x(D) + \text{Cauchy}\left(\frac{6\,\text{SS}^\beta(f_x, D)}{\epsilon}\right)$ is $(\epsilon, 0)$-differentially private.

We note that any upper bound on $\text{SS}^\beta(f_x, D)$ can be used in place of the true smooth sensitivity, as increasing the scale of the noise cannot degrade the privacy of the mechanism. By Theorem 5.1, if $f_x(D)$ is stable for some distance $k'$, then $\text{SS}^\beta(f_x, D) \leq e^{-\beta k'}$. Setting $\beta$ to its maximum value in the above mechanisms, and substituting our bound gives us that $f_x(D) + \text{Cauchy}\left(\frac{6\exp(-\epsilon k'/6)}{\epsilon}\right)$ satisfies $(\epsilon, 0)$-differential privacy. $\square$

## D.7. Proof of Theorem 5.3

*Proof.* Let $g$ be a non-private voting ensemble made up of $T$ binary classifiers $f^{(i)}$, $i = 1, \ldots, T$, each trained on disjoint subsets of the dataset $D$. We assume that each binary classifier prediction at the point $x$ is stable up to some distance $k^{(i)}$. We define the binary vote counts at a query point $x$ as

$$n_0(x) = |\{i : i \in \{1, \ldots, T\}, f^{(i)}(x) = 0\}| \tag{24}$$

$$n_1(x) = |\{i : i \in \{1, \ldots, T\}, f^{(i)}(x) = 1\}| \tag{25}$$

such that $n_0(x) + n_1(x) = T$. Let the response of the ensemble classifier trained on the dataset $D$ at the point $x$ be

$$g_x(D) = \begin{cases} 1 & \text{if } n_1(x) \geq n_0(x) \\ 0 & \text{if } n_1(x) < n_0(x) \end{cases} \tag{26}$$

We wish to lower bound the distance for which $g_x$ is stable with respect to additions or removals from the dataset $D$.

The number of votes that must be flipped to cause $g_x$ to change is given by $n = \left\lceil \frac{|n_1(x) - n_0(x)|}{2} \right\rceil$, i.e. at least half the distance between the vote counts. Now consider the number of additions or removals from $D$ that is required to cause at least $n$ votes to flip.

Since each individual classifier $f^{(i)}$ is stable at a distance of $k^{(i)}$, an adversary has to modify at least $K = \sum_{i \in S_n} k^{(i)}$ entries of $D$ to cause $n$ votes to flip. Here $S_n$ is the set of indices corresponding to the $n$ classifiers with the smallest stable distances $k^{(i)}$.

This suffices as a proof that $g_x$ is stable at a distance of $K$. Additionally, by Lemma 4.2, $g_x(D)$ being stable at a distance of $K$ implies

$$A^{K-1}(g_x, D) = 0. \tag{27}$$

$\square$

# E. Experimental Details and Further Results

In this section, we provide the full experimental details for the results in Section 6. All experiments are run on a server with 2x AMD EPYC 9334 CPUs and 2x NVIDIA L40 GPUs. Code to reproduce our experiments is available at [redacted for anonymity]. Where stated, all DP-SGD $(\epsilon, \delta)$ values are computed using the Opacus libary (Yousefpour et al., 2021).

For each dataset, we provide the number of epochs, learning rate $(\alpha)$, learning rate decay factor $(\eta)$, and batchsize $(b)$. Learning rate decay is applied using a standard learning rate schedule $\alpha_n = \alpha/(1 + \eta n)$.

### E.1. Blobs Dataset

In Figure 3, we examine a dataset consisting of 3,000 samples drawn from two distinct isotropic Gaussian distributions. The objective is to predict the distribution of origin for each sample in a supervised learning task. To highlight the dataset-dependent nature of our bounds, we vary both the cluster means and standard deviations. We train a single-layer neural network with 128 hidden neurons using AGT and visualise the resulting smooth sensitivity bounds. The model is trained for four epochs with hyperparameters set to $b = 3000$, $\alpha = 1.0$, $\eta = 0.6$, and $\gamma = 0.06$. A full set of bounds is computed for values of $k$ in $\mathcal{K} = \{1, \ldots, 100\}$. As discussed in Appendix C, running the complete range of $k$ values is unnecessary for achieving privacy gains; a carefully selected subset would be preferable in practical applications. Using these privacy bounds, we determine the maximum stable $k^\star \in \mathcal{K}$ for each point in a grid over the domain. This value is then used to compute a bound on the smooth sensitivity, which we present as a heat map.

In Figures 2, 4, and 5, our goal is to use the smooth sensitivity bounds computed above to improve private prediction in both the single model and ensemble settings. Here we use the "separable" blobs dataset with 5000 samples and train a logistic regression classifier. To facilitate tight bound propagation, we use the maximum batchsize of $b = 5000/T$ for each of the $T$ members of the ensemble and other hyperparameters the same as the above. We apply the global sensitivity or smooth sensitivity mechanisms to the same nominal models / ensembles.

### E.2. Retinal OCT Image Classification

Next, we consider another dataset with larger scale inputs: classification of medical images from the retinal OCT (OctMNIST) dataset of MEDMNIST (Yang et al., 2021). We consider binary classification over this dataset, where a model is tasked with predicting whether an image is normal or abnormal (the latter combines three distinct abnormal classes from the original dataset).

The model comprises two convolutional layers of 16 and 32 filters and an ensuing 100-node dense layer, corresponding to the 'small' architecture from (Gowal et al., 2018). To demonstrate our framework, we consider a base model pre-trained on public data and then fine-tuned on new, privacy-sensitive data, corresponding to the 7754 Drusen samples (a class of

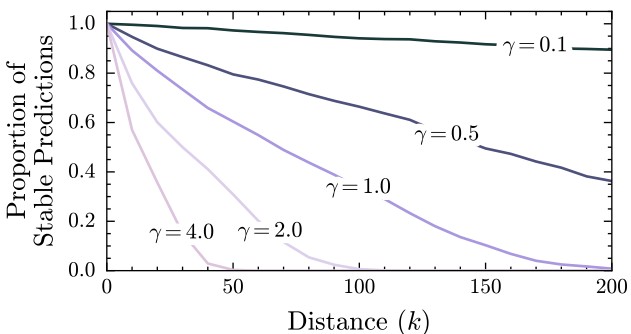 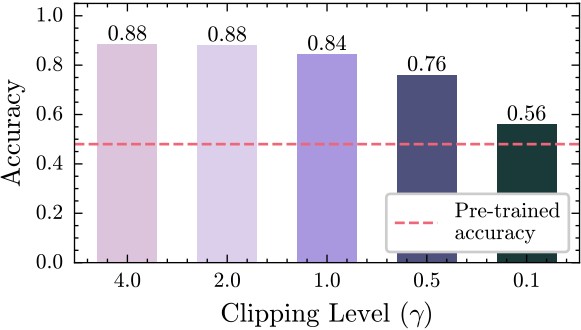

*Figure 7.* Performance of the fine-tuned classifier on the unseen diseased class (Drusen). Left: Proportion of the test set queries certified to be stable at a distance of $k$. Performance on new diseased class in OCT-MNIST Fine-Right: Prediction accuracy on the Drusen class.

abnormality omitted from initial training). First, we train the complete model excluding this using standard stochastic gradient descent. We then fine-tune only the dense layer weights to recognise the new class, with a mix of 50% Drusen samples per batch. We aim to ensure privacy only with respect to the fine-tuning data. The hyper-parameters used for fine-tuning using AGT are $E = 4, \alpha = 0.06, \eta = 0.5$; the batchsize is chosen to be the maximum possible for each ensemble size $T$.

Figure 7 shows the performance of the fine-tuned classifier on the (previously unseen) Drusen class. The pre-trained model achieves around 50% on the new disease (i.e. akin to random guessing). After fine-tuning, the accuracy on the Drusen data ranges from approximately 0.6 up to 0.88 after fine-tuning. The final utility is highly dependent on the value of the clipping level $\gamma$. Decreasing the value of $\gamma$ reduces the accuracy of the model, but increases the proportion of points for which we can provide certificates of stability. In our experiments in the main text, we choose a value of $\gamma = 1.0$, as a trade-off between utility and certification tightness. We note that for small values of $k$, our framework is able to provide certificates of prediction stability for well over 90% of test-set queries.

### E.3. IMDB Movie Reviews

Finally, we consider fine-tuning GPT-2 (Radford et al., 2019) for sentiment analysis on the large-scale (40,000 samples) IMDb movie review dataset (Maas et al., 2011). In this setup, we assume that GPT-2 was pre-trained on publicly available data, distinct from the data used for fine-tuning, which implies no privacy risk from the pre-trained embeddings themselves. Under this assumption, we begin by encoding each movie review into a 768-dimensional vector using GPT-2's embeddings.

We then train a fully connected neural network consisting of $1 \times 100$ nodes, to perform binary sentiment classification (positive vs negative reviews). Figure 8 shows how the nominal accuracy and the proportion of certified data points evolve over the course of training. Although the initial accuracy is equivalent to random chance, fine-tuning allows us to reach an accuracy close to 0.80, while simultaneously preserving

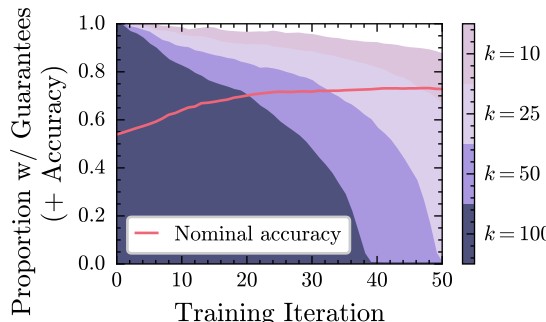

*Figure 8.* Proportion of test set queries certified to be stable at a distance of $k$ for the IMDB sentiment classification task.

strong privacy guarantees using AGT. Each training run of the sentiment classification model with AGT takes approximately 55 seconds, compared to 25 seconds when using standard, un-certified, training in pytorch. We note that our guarantees weaken with increased training time, indicating that stronger privacy guarantees can be obtained by terminating training early. In our experiments in the main text we choose to train with hyperparameters $E = 3, \alpha = 0.2, \eta = 0.5, \gamma = 0.04$, using the maximum possible batchsize available to each ensemble member.

### E.4. Additional Ablations: American Express Default Prediction

In order to understand the performance of Algorithm 1 in deeper networks, we turn to the American Express default prediction task. This tabular dataset[3] comprising 5.4 million total entries of real customer data asks models to predict whether a customer will default on their credit card debt. We train networks of varying depth with each layer having 128 hidden nodes. We highlight that fully connected neural networks are generally not competitive in these tasks, thus their accuracy is significantly below competitive entries in the competition. However, this massive real-world dataset in a privacy-critical domain enables us to test the scalability of our approach. In particular, we note in Section 4.3 that as batch size tends to infinity, our bounds become arbitrarily tight. On the other hand, since we employ interval bound propagation to bound the model gradients in Algorithm 1, our bounds weaken super-linearly as they propagate through deeper networks (Wicker et al., 2020).

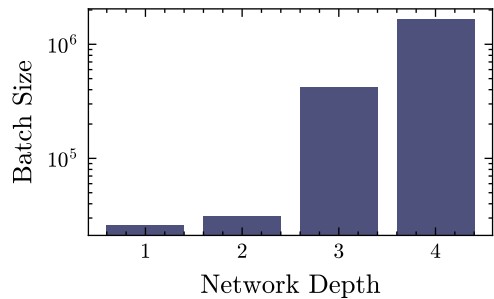

*Figure 9.* Minimum batch size required provide certificates of prediction stability at a distance of $k = 50$ for at least 95% of the AMEX test data.

Figure 9 shows the smallest batch sizes that allow us to train networks of varying depth with guarantees of prediction stability at a distance of $k = 50$ for at least 95% of test set inputs. Our approach requires a batch size of over one million to provide these guarantees for 4-layer neural networks. Ignoring the effect of this large batch size on performance for the sake of this particular case study, this example highlights that tightly bounding the model gradients, e.g., though bounds-tightening approaches (Sosnin & Tsay, 2024), will prove an important line of future research.

---

[3]See `www.kaggle.com/competitions/amex-default-prediction/`; accessed 05/2024

