# OpenReview forum: "Certification for Differentially Private Prediction in Gradient-Based Training"
_ICML.cc/2025/Conference — ICML 2025 poster_

### Official Review · Reviewer_g645 · 2025-03-13

**Overall Recommendation:** 4

**Summary:**

This paper presents a certification algorithm for assessing the stability of model predictions, which helps reduce the smooth sensitivity of the predictions. By providing a tighter bound on smooth predictions, the algorithm enhances the accuracy of private predictions. Empirical experiments demonstrate that this certification improves private binary classification and enhances the accuracy of noisy aggregation.

## update after rebuttal
I will keep my score unchanged.

**Claims And Evidence:**

The claims are valid. I appreciate how the authors introduce prediction stability—ensuring a stable prediction naturally leads to a reduction in smooth sensitivity. This insight effectively motivates the development of an algorithm to verify prediction stability.

**Essential References Not Discussed:**

No.

**Experimental Designs Or Analyses:**

Yes. The smooth sensitivity is reduced using the certification. Model prediciton accuracy is improved under certain cases.

**Methods And Evaluation Criteria:**

The experiments are well designed. The reduced smooth sensitivity helps improves performance on two application, private binary classification and PATE.

**Other Comments Or Suggestions:**

No.

**Other Strengths And Weaknesses:**

Strength: The use of certification for prediction stability to reduce smooth sensitivity is an innovative idea. The paper presents two important applications: private binary classification and PATE. Additionally, the writing is clear, and the motivations are well articulated.

Weakness: My primary concern is the efficiency of the algorithm. The certification process involves clipping (similar to DP-SGD) and incurs additional overhead for bounding stability. As the authors acknowledge, the algorithm requires 20 times more computation time. Given this substantial overhead, the improvement over DP-SGD is relatively modest, which may lead users to question whether adopting the new algorithm is worthwhile.

Another limitation, as stated in the paper, is that the current method is restricted to binary classification. Additionally, its performance in PATE is effective only when the number of queries is small.

**Questions For Authors:**

Could you also give the memory overhead for the Algorithm 1? For example, when running gpt2, what's the memory consumption compared to DP-SGD?

**Relation To Broader Scientific Literature:**

The paper gives an alternative way for private prediction with improved accuracy. This also motivates more investigation into private predictions which might be more accesiable compared to DP-SGD.

**Theoretical Claims:**

I didn't fully checked the proofs. But the results make sense to me.

---

> ### Author Rebuttal · Authors · 2025-04-01
>
> We thank the reviewer for their time and consideration of our work.
>
>
> * Could you also give the memory overhead for the Algorithm 1? For example, when running gpt2, what's the memory consumption compared to DP-SGD?
>
> Both DP-SGD and AGT require the computation of per-sample gradients, incurring a large memory overhead compared to standard pytorch training. However, much like DP-SGD, AGT can employ “virtual batches” to separate physical steps (gradient computation) and logical steps (parameter updates) (see https://opacus.ai/docs/faq, for example). This means that memory overhead can be managed by only storing per-sample gradients for a subset of a batch at a time. Computing SEMin/SEMax independently over each virtual batch amounts to storing the top-k per-sample gradients, incurring a memory overhead of O(k * no. parameters) during training. We thank the reviewer for the questions and will reference this in a revised version of our main text or if there is not space to do so will have a discussion of this in our Appendix.
>
> * Given this substantial overhead, the improvement over DP-SGD is relatively modest, which may lead users to question whether adopting the new algorithm is worthwhile.
>
> We first point out that the computational cost of our approach actually compares favorably with other private prediction methods that use ensembles. When compared with DP-SGD we do acknowledge that it requires additional overheads, but the privacy analyses are not exactly comparable. Additionally, as we point out in our global response, there are many avenues for future works to sharpen the bounds that we provide using the current instantiation of our framework. With many future directions for improvement we hope that this line of research yields privacy guarantees that are competitive with DP-SGD.
>
> * Another limitation, as stated in the paper, is that the current method is restricted to binary classification.
>
> We agree with the reviewer that our current practical instantiation of the framework is only practically validated on binary classification, however, the framework itself is general and future works will be able to extend this to regression and multi-class classification by leveraging tighter or use-case specific bounds.
>
> * Its performance in PATE is effective only when the number of queries is small.
>
> We would like to highlight that in our initial submission we do not use our bounds directly in the student-teacher set up of PATE, but only compare the mechanisms in the context of private prediction. However, as indicated in our response to reviewer dcek, using our private prediction mechanism to train a student allows the privacy budget to be fixed for an unlimited number of future queries.
>
> With regards to ensemble approaches to private prediction, however, it is necessarily true that as the number of queries grows so does the expended privacy budget. However, we emphasize that the tighter privacy bounds of our approach are an orthogonal development to other private prediction methods and in theory can tighten other approaches enabling us to answer more queries within the same budget. For example, if the ensemble is using the global sensitivity in order to privatize its predictions, then there are cases (as we show in our paper) where our bound makes things strictly tighter thus allowing for more queries to be answered at the same privacy budget. We thank the reviewer for the comment and will try to clarify this in future works.

---

### Official Review · Reviewer_VLat · 2025-03-14

**Overall Recommendation:** 4

**Summary:**

This paper introduces a new approach for improving differential privacy in machine learning predictions. The authors propose a method to compute tighter dataset-specific upper bounds on prediction sensitivity by using convex relaxation and bound propagation techniques. Their approach called abstract gradient training analyses how model parameters change when data points are added or removed from the training set. By combining these bounds with smooth sensitivity mechanisms, they achieve significantly better privacy-utility trade-offs compared to methods based on global sensitivity. The authors evaluate their approach on medical images and sentiment analysis. The method allows users to dynamically adjust privacy budgets and works with complex training configurations like federated learning.

## Update after rebuttal

I originally recommended accept and had some non-urgent comments. The authors responded well to those and I kept my score (4).

**Claims And Evidence:**

- The authors claim that fewer than 10 runs of the AGT algorithm are typically sufficient to capture most privacy benefits. Although preliminary experimental results support this claim for the chosen tasks, the paper lacks a more systematic sensitivity analysis. The robustness of this claim in more diverse settings is not explored.
- The computational overhead (reported as 20–40× standard training) may limit the practicality claims of the method despite the improved sensitivity bounds.

**Essential References Not Discussed:**

There are a few papers that I feel the authors might have missed in this area, [1] for the background of smoothness in privacy and [2] for inclusion in the discussion about inaccurate estimation in privacy.


[1] Nissim, Kobbi, Sofya Raskhodnikova, and Adam Smith. "Smooth sensitivity and sampling in private data analysis." Proceedings of the thirty-ninth annual ACM symposium on Theory of computing. 2007.

[2] Casacuberta, Sílvia, et al. "Widespread underestimation of sensitivity in differentially private libraries and how to fix it." Proceedings of the 2022 ACM SIGSAC Conference on Computer and Communications Security. 2022.

**Experimental Designs Or Analyses:**

While the experiments compare the proposed method against baselines across various privacy budgets and even examine ensemble size effects, the discussion of hyperparameter tuning is relatively brief. It is not entirely clear how sensitive the performance is to choices like batch size, number of epochs, or the precise method used for bound propagation.

**Methods And Evaluation Criteria:**

- The authors test on both imaging and natural language data, which demonstrates the versatility of the approach.
- However, the experimental validation is limited to a few binary tasks. It remains unclear whether these tighter bounds extend practically to more complex models (e.g. deep multi-class networks) or larger scale real world applications. That remains more of a theoretical promise.

**Other Comments Or Suggestions:**

- The operators SEMin and SEMax are only briefly described, a clearer definition would be better
- PATE is also mentioned without description or definition

**Other Strengths And Weaknesses:**

Strengths:
- The authors present a novel approach that bridges verification-based techniques with differential privacy theory
- The work provides rigorous mathematical proofs and analysis throughout
- The authors test their approach across various datasets and model architectures

Weaknesses:
- The method’s performance is sensitive to hyperparameter choices such as batch size and the number of training epochs. These dependencies, along with the assumption of a fixed data ordering, might restrict the practical applicability of the method in more dynamic training scenarios.
- The method's effectiveness diminishes as datasets become less separable or when limited data is available, so potential fragility under challenging data conditions.
- Some of the technical details of bound propagation and sensitivity certification might benefit from additional explanation but this is not a requirement.

**Questions For Authors:**

1. Might the element wise clipping have a different bias-variance tradeoff compared to the more standard l2 clipping? What effect could this have
2. 40x overhead would be impractical for many real world applications, do the authors see any avenues or potential for reducing this?
3. Cauchy noice has been used in some previous work, but do the authors see any specific advantages over Laplace noise?
4. Could it be possible to quantify or characterise what level of data separability is needed for the method to provide meaningful advantages over global sensitivity approaches?
5. Have any automatic approached been explored for selecting optimal k values that might reduce the need for multiple training runs while still achieving tight bounds?

**Relation To Broader Scientific Literature:**

Overall, the paper provides a reasonably good discussion of how its contributions build on and extend prior work in both differential privacy and neural network verification. But a few areas could benefit from deeper contextualization. For example, more explicit comparisons with the smooth sensitivity framework introduced by Nissim et al. (2007) and with recent advances in multi-neuron convex relaxations for neural network verification e.g. Wong and Kolter (2018) and Tjandraatmadja et al. (2020)

**Theoretical Claims:**

- The method development is thorough and mathematically grounded. However, many proofs are deferred to appendices with some key assumptions.
- interval bound propagation (IBP) to compute gradient bounds, while innovative, may be sensitive to the choice of activation functions and loss functions.
- Algorithm 1 is grounded, some proofs are deferred to the appendix which I checked and did not see any issues with

---

> ### Author Rebuttal · Authors · 2025-04-01
>
> We thank the reviewer for their time and careful review of our work.
>
> * The authors claim that fewer than 10 runs of the AGT algorithm are ... sufficient ... [this] lacks a more systematic sensitivity analysis.
>
> We appreciate the reviewer's point that, though empirically we find a small number of AGT runs is sufficient and we believe this is a general result, we cannot rule out the need for many runs of AGT. We would like to stress that in Appendix C we explicitly study how different numbers of AGT runs result in tighter bounds, which is in fact an attempt to study the robustness of this claim. As this message may have been unclear, in our revision we will more explicitly reference this section to the main text.
>
> * The computational overhead (reported as 20–40× standard training) may limit the practicality ... any avenues or potential for reducing this?
>
> The computational overhead of our method as presented is indeed a practical hurdle of its adoption; however, we note that this paper is the first instantiation of a new approach to private predictions and we hope that -- as with other approaches to privacy -- future research will improve our bounds and reduce this overhead.
>
> To reduce this we highlight the tightness of our smooth sensitivity bounds depends on the tightness of the local sensitivity bound and the number of values of k at which AGT is run for. Given sufficiently tight bounds on the local sensitivity (e.g., with future advancements), running AGT for even a single value of k may be sufficient to realise significant privacy benefits. Tighter bounds and careful choice of k values will be a direction for future work. Similarly, particular models may have tighter local sensitivity bounds (i.e., particular architecture or learning choices) may also be valuable directions of study.
>
> * Interval bound propagation (IBP) ... may be sensitive to the choice of activation functions and loss functions.
>
> One benefit of our proposed approach is that any bound propagation technique (e.g., one chosen to match the model) can be used to tighten our bounds. his will also be a future direction of work to tighten bounds in particular cases.
>
> * It is not entirely clear how sensitive the performance is to choices like batch size, number of epochs
>
> In the current version of our submission we highlight that batch size is typically taken to be the maximum possible size as this results in the tightest bounds. In a revised version we will include an ablation of the batch size in Appendix E.
>
> * More explicit comparisons with the smooth sensitivity framework introduced by Nissim et al. (2007), ... Wong and Kolter (2018) ...Tjandraatmadja et al. (2020)
>
> Our approach to tightened privacy explicitly uses the framework of Nissim et al. (2007); however, adapting our approach to further developments in local DP is an important future work. With regards to further convex relaxations, our framework is general and can use any propagation method. We thank the reviewer and will clarify this and we add references to the works the reviewer mentions.
>
> * There are a few papers that I feel the authors might have missed in this area, [1] ... and [2].
>
> We do have citation of [1], however, we are aware that there are multiple versions online, though each has the same theoretical framework that we reference and use extensively. We did not cite [2] and will do so in a revision of our submission.
>
> * Might the element wise clipping have a different bias-variance tradeoff ...?
>
> Yes, the element-wise clipping may introduce different biases in the final model. The preliminary results in this paper, particularly Figure 5, highlight that we actually observe better utility than DP-SGD indicating that this is not substantial.
>
> * Cauchy noice has been used ... advantages over Laplace noise?
>
> Smooth sensitivity requires the use of an “admissible” noise distribution, each of which comes with its own theoretical privacy loss. Of these choices discussed by Nissim et al., only Cauchy noise satisfies pure (delta=0) differential privacy.
>
> * ... quantify or characterise what level of data separability ... to provide meaningful advantages over global sensitivity approaches?
>
> We thank the reviewer for the questions as it raises a very interesting point. We present Figure 3 which visualizes this phenomenon and in our GPT experiment we hypothesize that separability is the reason for the strong bounds. However, characterizing separability itself in a general way is non-trivial and therefore characterizing the relationship between our bounds and separability is also non-trivial and an interesting future work.
>
> * Have any automatic approached been explored for selecting optimal k values ...?
>
> Selecting optimal k values is a challenging problem with many potential heuristic approaches that may be effective in reducing the computational overhead of our method and may be explored in future works.

---

> > ### Comment · Reviewer_VLat · 2025-04-02
> >
> > I thank the authors for their rebuttal which adequately addressed my concerns. The clarifications regarding AGT sensitivity analysis, use of Cauchy noise and separability considerations are particularly helpful. With the promised additions the paper will be strengthened. I maintain my position that this is a solid contribution and recommend acceptance.

---

### Official Review · Reviewer_BNAf · 2025-03-14

**Overall Recommendation:** 4

**Summary:**

This paper studies upper bounds on the sensitivity of prediction in machine learning models. By doing that, the paper presents tighter privacy analysis. After which, experimental results showing a wide improvement in the tightness of the privacy bounds.
## update after rebuttal
I raised my score to a 4.

**Claims And Evidence:**

The claims are all clear and convincing.

**Essential References Not Discussed:**

N/A

**Experimental Designs Or Analyses:**

The experiments done are extensive and support the result of the paper. The experiments are sound.

**Methods And Evaluation Criteria:**

The paper uses on multiple datasets, and the results show a large improvement in tightening the privacy predictions.

**Other Comments Or Suggestions:**

N/A

**Other Strengths And Weaknesses:**

The paper is well written and easy to read. The methods used are not the most novel, but the work has significance.

**Questions For Authors:**

N/A

**Relation To Broader Scientific Literature:**

Privacy is a very important topic in real world scenarios, and offering tighter bounds and privacy prediction stability moves us a step closer to being able to deploy DP in more applications.

**Theoretical Claims:**

I checked the proofs and discussions in both the main paper and the appendices, and to the best of my knowledge, there are no issues.

---

> ### Author Rebuttal · Authors · 2025-04-01
>
> We thank the reviewer for their kind words and for the careful work of checking the proof and technical steps of our work. In their review they did not necessarily provide a strong signal of the weaknesses they would like to see addressed in relation to their score. We hope that both our response to other reviewers and promised updates that we have address any concerns that they may have such that they are confident in recommending acceptance.

---

### Official Review · Reviewer_dcek · 2025-03-16

**Overall Recommendation:** 3

**Summary:**

The paper proposes to bound local sensitivity of predictions of models learned with gradient-based methods using interval bound propagation. Further, the paper uses the result to construct a sample-and-aggregate procedure for prediction ensembles. The paper then demonstrates that using the proposed bounds enables to significantly improve utility for the same query budget over global sensitivity.

## Update after rebuttal

I strongly suggest to quantitatively compare the data-dependent bounds obtained with IBP and the standard data-dependent bounds for report-noisy-max from Papernot et al., 2016, 2017, key baseline data-dependent analyses not compared to in the current version.

**Claims And Evidence:**

The claims are well supported by theory and experimental evaluation.

**Essential References Not Discussed:**

The [2018 version of PATE](https://arxiv.org/pdf/1802.08908) also uses smooth sensitivity. To be completely well-positioned with respect to the prior work on private prediction in machine learning, the paper should (1) compare the IBP bounds to PATE data-dependent bounds, and (2) have a new experiment comparing student-teacher pipeline when using the proposed approach and the PATE approach.

**Experimental Designs Or Analyses:**

See the "Methods and Evaluation Criteria" section.

**Methods And Evaluation Criteria:**

The experimental settings in the paper make sense to show that the proposed local sensitivity framework outperforms global sensitivity in single-model private prediction and ensemble prediction.

Considering that PATE is mentioned several times as a motivation for the ensemble setting, it is quite strange to not see a comparison with PATE. Indeed, this looks like one additional step of training student models on top of the experiment in Fig. 5. Moreover, PATE  also relies on smooth sensitivity (specifically, the [2018 version](https://arxiv.org/abs/1802.08908)) and data-dependent privacy bounds. The present paper does not seem to compare the bounds obtained with IBP to these standard simpler bounds, only comparing with global sensitivity. This seems to be a significant drawback in the evaluation setting. I would be happy to increase my score if such results were available.

**Other Comments Or Suggestions:**

- L693: GTP -> GPT
- Fig 2 and 3 have a different order than referenced in the text.

**Other Strengths And Weaknesses:**

Connecting the literature on adversarial robustness certification and DP is valuable, and might be an avenue for a fruitful future direction of research.

**Questions For Authors:**

---

**Relation To Broader Scientific Literature:**

The paper introduces a new way of computing local sensitivity applicable to machine learning by using interval bound propagation. This establishes a new intersection between adversarial robustness certification and differential privacy.

**Theoretical Claims:**

To the best of my understanding, the claimed results seem correct, given the IBP propagation bounds are correctly applied.

---

> ### Author Rebuttal · Authors · 2025-04-01
>
> We thank the reviewer for their time and consideration of our work.
>
> * It is quite strange to not see a comparison with PATE. Indeed, this looks like one additional step of training student models on top of the experiment in Fig. 5.
>
> Fig. 4 illustrates the privacy-utility tradeoff of our method compared to the subsample and aggregate mechanism employed by PATE. As a result, any gains in tightness of privacy analysis here directly translate into the privacy-utility costs of training subsequent student models using the PATE framework. While the comparison with DP-SGD in Fig. 5 would be more favourable following a student-teacher framework, this trade-off has been well studied in previous works. Further, this would require consideration of modified training settings, e.g. having access to an unlabelled public dataset. Nonetheless, we agree with the reviewer that understanding how the gains in our approach translate to improved trade-offs in student models is an interesting and potentially valuable future contribution. In this direction, we provide some preliminary results below, that will accompany a more thorough discussion in our appendix.
>
> * I would be happy to increase my score if such results were available.
>
> In a revised version of the paper, we will incorporate this discussion into the main text and reference an appendix section that presents preliminary results on applying our mechanism in a teacher-student training setting. Specifically, we re-run the experimental setup from Figure 5, using a privacy budget of (\epsilon, \delta) = (10, 10^{-5}) to label Q = 100 data points held out from the training dataset (which we assume to be our "public" unlabeled dataset). We emphasize that training a student model does fix the privacy budget and which privacy budget is selected will have a significant effect on the results. Once this is done, resulting teacher-generated labels are then used to train a student model. Our findings indicate that the student model's performance under each mechanism aligns with the accuracy levels observed at the corresponding inference budget in Figure 5. We hope this increases the reviewer's confidence in our contribution.
>
> | Teacher Mechanism                           | Blobs | OctMNIST | IMDB |
> | ------------------------------------------- | ----- | -------- | ---- |
> | Single model, global sensitivity            | 82.8  | 12.7     | 54.4 |
> | Single model, smooth sensitivity            | 99.8  | 18.7     | 73.5 |
> | Subsample and aggregate, global sensitivity | 99.5  | 14.1     | 73.0 |
> | Subsample and aggregate, smooth sensitivity | 98.1  | 19.8     | 71.7 |
> | DP-SGD                                      | 1.0   | 81.2     | 70.5 |
>
>
> * Moreover, PATE also relies on smooth sensitivity (specifically, the 2018 version) and data-dependent privacy bounds.
>
> We thank the reviewer for highlighting this variant of PATE that leverages tighter, data-dependent privacy analysis. We will be sure to cite it in the revised version of our submission. While we currently do not include comparisons with such methods, we highlight to the reviewer that the tightening approach introduced in our work is orthogonal to those used in subsequently developed PATE mechanisms. We emphasize that this suggests that our bounds could potentially be combined with these tighter privacy analyses to enable even stronger privacy guarantees. We are grateful for this insightful suggestion and will make it clear in our revision that more advanced mechanisms exist beyond those we evaluate, and that integrating our approach with them represents a promising direction for future research.

---

> > ### Comment · Reviewer_dcek · 2025-04-05
> >
> > Thank you for the response.
> >
> > For "I would be happy to increase my score if such results were available" the results I also meant the comparison to the existing data-dependent bounds.
> >
> > Let me make this more precise, and let us forget the 2018 PATE for the moment. Consider Theorem 3 in the original PATE paper. It provides a data-dependent privacy guarantee of report noisy max with Laplace noise used when aggregating predictions from a teacher model ensemble (the 2018 version of PATE introduces additional results on this). As far as I understand, this is directly an appropriate baseline for your new method based on IBP, at least in  some of the settings. Could you please explain in detail why is it orthogonal? If this is in fact not orthogonal, then could you provide a comparison of, e.g. obtained epsilon values with the classical PATE data-dependent analysis and the proposed IBP-based analysis?

---

> > > ### Author Response · Authors · 2025-04-09
> > >
> > > We thank the reviewer for clarifying their statement. We agree that our current comparison for ensemble models focuses on the basic privacy analysis presented in the original PATE paper (Papernot et al., 2016), whereas that work also includes a tighter, data-dependent privacy accounting. The data dependence in their analysis arises from usage of the vote histogram corresponding to specific predictions. In contrast, our method incorporates the sensitivities of the models themselves to the training data, as well as the vote histogram. The tighter privacy analysis from the original PATE paper indeed represents an interesting and relevant baseline, and we will aim to include corresponding results in the final version of the paper.

---

### Decision · Program_Chairs · 2025-05-01

**Decision:**

Accept (poster)

**Comment:**

The reviewers collectively recognize the paper's valuable contribution in proposing tighter data-dependent privacy bounds using interval bound propagation for gradient-based learning models. The theoretical analysis is thorough, rigorous, and successfully bridges adversarial robustness certification techniques with differential privacy, a promising interdisciplinary direction. Despite noted concerns about computational overhead and limited experimental scope, the authors' rebuttal effectively addressed these points, clarifying future directions and providing additional experimental validations. Overall, the reviewers were positive about the paper.